# Giantin Is Required for Post-Alcohol Recovery of Golgi in Liver Cells

**DOI:** 10.3390/biom8040150

**Published:** 2018-11-16

**Authors:** Carol A. Casey, Paul Thomes, Sonia Manca, Armen Petrosyan

**Affiliations:** 1Department of Internal Medicine and VA-Nebraska-Western Iowa Health Care System, Omaha, NE 68105, USA; ccasey@unmc.edu (C.A.C.); paul.thomes@unmc.edu (P.T.); 2Department of Biochemistry and Molecular Biology, College of Medicine, University of Nebraska Medical Center, Omaha, NE 68198, USA; sonia.manca@unmc.edu; 3The Nebraska Center for Integrated Biomolecular Communication, Lincoln, NE 68588, USA; 4The Fred and Pamela Buffett Cancer Center, Omaha, NE 68106, USA

**Keywords:** alcohol-induced Golgi disorganization, Golgi recovery, giantin, hepatic proteins, ethanol withdrawal

## Abstract

In hepatocytes and alcohol-metabolizing cultured cells, Golgi undergoes ethanol (EtOH)-induced disorganization. Perinuclear and organized Golgi is important in liver homeostasis, but how the Golgi remains intact is unknown. Work from our laboratories showed that EtOH-altered cellular function could be reversed after alcohol removal; we wanted to determine whether this recovery would apply to Golgi. We used alcohol-metabolizing HepG2 (VA-13) cells (cultured with or without EtOH for 72 h) and rat hepatocytes (control and EtOH-fed (Lieber–DeCarli diet)). For recovery, EtOH was removed and replenished with control medium (48 h for VA-13 cells) or control diet (10 days for rats). Results: EtOH-induced Golgi disassembly was associated with de-dimerization of the largest Golgi matrix protein giantin, along with impaired transport of selected hepatic proteins. After recovery from EtOH, Golgi regained their compact structure, and alterations in giantin and protein transport were restored. In VA-13 cells, when we knocked down giantin, Rab6a GTPase or non-muscle myosin IIB, minimal changes were observed in control conditions, but post-EtOH recovery was impaired. Conclusions: These data provide a link between Golgi organization and plasma membrane protein expression and identify several proteins whose expression is important to maintain Golgi structure during the recovery phase after EtOH administration.

## 1. Introduction

The Golgi apparatus is the central sorting and transportation hub involved in the posttranslational modification and sorting of cargo molecules, and delivering them to appropriate cellular locations or to the exocytic and endocytic pathways [1]. In mammalian cells, Golgi is a highly organized, perinuclear, ribbon-like structure, composed of stacks of flattened and elongated cisternae. In hepatocytes, Golgi coordinates glycosylation and trafficking of different glycoproteins that play important roles in the secretory and detoxification function of the liver. 

In response to stress, including exposure of cytotoxic agents, the compact Golgi structure can undergo remodeling characterized by varying degrees of scattering and unstacking [2,3,4,5,6,7]. Multiple studies have demonstrated that in hepatocytes, ethanol (EtOH) administration and subsequent metabolism can alter the structure of the Golgi [8,9,10]. Importantly, such fragments of Golgi are capable of eliciting antibody production, and anti-Golgi antibodies are markedly elevated in the sera of end-stage liver disease induced by heavy alcohol consumption, underscoring the clinical relevance of Golgi fragmentation in alcohol-induced organ dysfunction [11]. One protein, giantin, appears to be especially important for Golgi’s compact structure. Giantin is the highest molecular weight (376 kDa) Golgi matrix protein. It consists of a short C-terminal domain located in the Golgi lumen [12], where a disulfide bond connects two monomers to form an active homodimer, which is followed by a one-pass trans-membrane domain and then a large (≥350 kDa) N-terminal region projecting into the cytoplasm. This unique structure suggests that giantin is the core Golgi protein and therefore could be essential for cross-bridging cisternae during Golgi biogenesis [13]. Giantin dimerization appears to be catalyzed by the chaperone, protein disulfide isomerase A3 (PDIA3) [13,14,15]. This chaperone is carried by the coat protein complex II (COPII) vesicles, of which SAR1A GTPase is an essential component. We have recently observed that a key event in EtOH-induced Golgi disorganization is the inactivation of the SAR1A GTPase [16]. Given the significant role played by giantin dimers in maintaining Golgi structure [15,17,18], the inactivation of SAR1A could result in a lack of giantin in Golgi membranes and subsequent Golgi disorganization. 

Alterations in Golgi morphology appear to be accompanied by the impaired trafficking and secretion of several essential hepatic glycoproteins. For example, transferrin was found to be retained in the endoplasmic reticulum (ER) and Golgi of the hepatocytes in both human alcoholic liver cirrhosis and in livers of rats fed with EtOH, causing impairment of its iron transport function [19]. Similarly, in cellular and rat models of chronic alcohol exposure, we observed the deposition of asialoglycoprotein receptor (ASGP-R) in *cis*- and *medial*-Golgi [16]. In addition to this, the activities of different glycosyltransferases are reduced in both ER and Golgi after EtOH administration [20,21]. Some of these Golgi resident enzymes exhibit altered re-localization due to EtOH-induced impairment of COPI vesicles, which normally deliver these enzymes to appropriate sites within the Golgi [22,23]. Also, recently we and others found that giantin represents a Golgi docking site for different Golgi resident proteins [15,24,25,26], and EtOH-induced alteration of giantin dimerization results in impaired Golgi targeting of mannosyl (α-1,3)-glycoprotein beta-1,2-*N*-acetylglucosaminyltransferase (MGAT1), the key enzyme of *N*-glycosylation [22]. This, in turn, causes abnormal glycosylation of ASGP-R and could be a potential reason for the release of carbohydrate-deficient transferrin, the widely available test for determining recent alcohol consumption [27]. Overall, alcohol-induced Golgi fragmentation has a significant impact on protein homeostasis in hepatocytes and could play a crucial role in the development of alcoholic liver disease (ALD), which remains a major cause of liver-related mortality in the US and worldwide [28]. 

It is known that abstinence is the most important therapeutic intervention for patients with ALD [29,30,31]. Also, it is been observed that periodic drinking is associated with a lower risk of ALD than daily drinking [32], implying the ability of organelles to recover during alcohol-free days [33,34]. Additionally, previous work from our laboratory and others has shown that that alcohol-induced impairments in morphology, injury, and protein trafficking can be reversed after removal of alcohol [35]. Since Golgi has a remarkable self-organizing mechanism [36], and in most cases, Golgi returns to its classical structure and positioning as soon as cells return to a drug- or stress-free condition [37,38,39,40], we hypothesized that these organelles would also show signs of recovery after ethanol withdrawal, and the aim of our study was to analyze the mechanism of Golgi recovery during alcohol abstinence. Using both a rat model and the recombinant HepG2 (VA-13) cells that efficiently express hepatic alcohol dehydrogenase (ADH) [41], we found that recovery of Golgi after EtOH withdrawal is associated with re-dimerization of giantin. Further, giantin is required for post-alcohol recovery of Golgi and the consequent trafficking of hepatic proteins. Additionally, restoration of Golgi morphology requires active Rab6a GTPase and action of the non-muscle myosin IIB motor protein. 

## 2. Materials and Methods

### 2.1. Antibodies and Reagents

The primary antibodies used were: (a) rabbit polyclonal—giantin (Novus Biologicals, Centennial, CO, USA, NBP2-22321), giantin (Abcam, Cambridge, United Kingdom, ab24586 and ab93281), ASGP-R1 (Abcam, ab88042), NMIIA (Abcam, ab75590), polymeric immunoglobulin receptor (PIGR) (Abcam, ab96196), transferrin (Dako, Carpinteria, CA, USA, A0061); (b) rabbit monoclonal—Golgin subfamily A member 2 (GM130) (Abcam, ab52649), Golgi reassembly-stacking protein of 65 kDa (GRASP65) (Abcam, ab174834); (c) mouse monoclonal—NMIIB (Abcam, ab684), GRASP65 (Santa Cruz Biotechnology, Dallas, TX, USA, sc365434), β-actin (Sigma-Aldrich, St. Louis, MO, USA, 2228), giantin (Abcam, ab37266), glyceraldehyde-3-phosphate dehydrogenase (GAPDH; Sigma-Aldrich, G8795); (d) mouse polyclonal—GM130 (Abcam, ab169276). The secondary antibodies (Jackson ImmunoResearch, West Grove, PA, USA) were: (a) horseradish peroxidase (HRP)-conjugated donkey anti-rabbit (711-035-152) and donkey anti-mouse (715-035-151) for Western blot (W-B); (b) donkey anti-mouse Alexa Fluor 488 (715-545-150) and anti-rabbit Alexa Fluor 594 (711-585-152) for immunofluorescence. 

### 2.2. Cell Culture, Ethanol Administration, and Isolation of Rat Hepatocytes

HepG2 cells transfected with mouse ADH1 (VA-13 cells) were obtained from Dr. Dahn Clemens at the Department of Veterans Affairs, Nebraska Western Iowa Health Care System, US [41]. VA-13 cells were grown in Dulbecco’s modified Eagle medium (DMEM) with 4.5 g/mL glucose, 10% fetal bovine serum (FBS), non-essential amino acids and 100 U/mL of Penicillin-Streptomycin. Twenty-four hours after seeding cells (at ∼75% confluence), culture media were changed for one containing 35 mM EtOH for another 72 h. The medium was replaced every 12 h to maintain a constant EtOH concentration. Control cells were seeded at the same time as treated cells and maintained in the same medium; EtOH was replaced by the appropriate volume of medium to maintain similar caloric content. For post-alcohol recovery, after EtOH treatment cells were maintained in regular medium for another 48 h. In another series of experiments, cells were treated with 35 mM EtOH for 72 h, then these cells were transfected with 150 nM giantin small interfering RNA (siRNA) followed by incubation in regular medium for 48 h. 

Primary rat hepatocytes from control and EtOH-fed animals were prepared from male Wistar rats. Rats weighing 140–160 g were purchased from Charles River Laboratories (Wilmington, MA, USA). Initially, animals were fed a Purina chow diet and allowed to acclimate to their surroundings for a period of 3 days. Then rats were paired according to weight and fed either control or EtOH containing (35% fat, 18% protein, 11% carbohydrates, 36% EtOH) Lieber-DeCarli diet for 5 weeks (Dyets, Inc., Bethlehem, PA, USA). The control diet was identical to the EtOH diet, except for the isocaloric substitution of EtOH with carbohydrates [42]. This protocol was approved by the Institutional Animal Care and Use Committee of the Department of Veterans Affairs, Nebraska Western Iowa HCS, and the University of Nebraska Medical Center (Protocol no. 14-007-03-FC). For recovery, rats were administered the control diet for 10 days. Hepatocytes were obtained from livers of control and EtOH-fed rats by a modified collagenase perfusion technique as described and used previously by the Casey laboratory [43,44]. The primary hepatocytes isolated from control and EtOH-treated animals were cultured, as previously described [45]. Briefly, freshly isolated hepatocytes were seeded in William’s media on collagen-coated six-well plates with coverslips. After 2 h in culture, cells were washed with phosphate-buffered saline (PBS), followed by incubation with 5% FBS-Williams media. Cells were maintained at 37 °C in 5% CO_2_ for the indicated time. Additional cell aliquots were washed in cold PBS, and the pellets were stored at −70 °C for future analysis.

### 2.3. Human Liver Tissues

De-identified normal and alcoholic cirrhotic frozen liver tissues were obtained from the Liver Tissue Cell Distribution System (LTCDS), Minneapolis, MN, USA, funded by NIH Contract number no. HSN276201200017C. Liver tissues were stored at −70 °C until analysis. A portion of the liver tissue was homogenized in 0.05 M Tris-HCl, 0.25 M Sucrose (pH 7.4) supplemented with protease and phosphatase inhibitors (Sigma-Aldrich), and centrifuged at 1000× *g* to obtain postnuclear supernatant (PNS), as previously described [46]. Proteins from freshly made PNS were subjected to Western blotting for detection of giantin dimerization as described in figure legends. 

### 2.4. Immunoprecipitation and Transfection

For identification of proteins in the complexes pulled down by immunoprecipitation (IP), confluent cells grown in a T75 flask were washed three times with 6 mL PBS each, harvested by trypsinization, and neutralized with soybean trypsin inhibitor (Sigma-Aldrich) at a 2× weight of trypsin. Immunoprecipitation steps were performed using Pierce Co-Immunoprecipitation Kit (Thermo Fisher Scientific) according to the manufacturer instructions. Mouse and rabbit non-specific IgG was used as non-specific controls. All cell lysate samples for IP experiments were normalized by appropriate proteins. To determine whether the target protein was loaded evenly, input samples were preliminarily run on a separate gel with different dilutions of control samples vs. treated, then probed with anti-target protein antibodies. The intensity of obtained bands was analyzed by ImageJ software version 1.52h (National Institutes of Health (NIH), USA), and samples with identical intensity were subjected to IP. MYH9 (myosin, heavy polypeptide 9, non-muscle, NMIIA), MYH10 (myosin, heavy polypeptide 10, non-muscle, NMIIB), GOLGB1 (giantin), GOLGA2 (GM130), GORASP1 (GRASP65), Rab6a, and scrambled on-target plus smartpool siRNAs were purchased from Santa Cruz Biotechnology. All products consisted of pools of three target-specific siRNAs. Cells were transfected with 100–150 nM siRNAs using Lipofectamine RNAi MAX reagent (Thermo Fisher Scientific, Waltham, MA, USA) PCMV-intron myc Rab6 T27N was a gift from Terry Hebert (Addgene plasmid #6782) [47]. Transient transfection of cells was carried out using Lipofectamine 3000 (Thermo Fisher Scientific, Waltham, MA, USA) following manufacturer protocol. 

### 2.5. Confocal Immunofluorescence Microscopy

Staining of cells was performed by methods described previously [48]. Slides were examined under a Zeiss 510 Meta Confocal Laser Scanning Microscope and LSM 800 Zeiss Airyscan (Carl Zeiss AG, Oberkochen, Germany) microscope performed at the Advanced Microscopy Core Facility of the University of Nebraska Medical Center. Images were analyzed using ZEN 2009 Light Edition 5.5.285 software (Carl Zeiss AG). For some figures, image analysis was performed using Adobe Photoshop (Adobe Inc. San Jose, CA, USA) and ImageJ. Assessment of fragmented Golgi versus perinuclear was performed as previously described [48].

### 2.6. Plasma Membrane Protein Isolation and Glycan Assessment

Plasma membranes were isolated using Pierce Chemical kit (Thermo Scientific) according to their protocol. To analyze glycosylation of proteins from plasma membrane fraction, samples were run on 10% SDS-PAGE followed by W-B with HRP-conjugated *Sambucus nigra* lectin (SNA), which binds preferentially to sialic acid attached to terminal galactose in α-2,6 and to a lesser degree, α-2,3 linkage.

### 2.7. Statistical Analysis

Data are expressed as mean ± SD. The analysis was performed using a 2-sided *t*-test in Excel (Microsoft Corporation, Redmond, WA, USA). A value of *p* < 0.05 was considered statistically significant.

### 2.8. Miscellaneous

Protein concentrations were determined with the Coomassie Plus Protein Assay (Pierce Chemical Co., Rockford, IL, USA) using bovine serum album (BSA) as the standard. Densitometric analysis of band intensity was performed using ImageJ.

## 3. Results

### 3.1. Structure of Golgi and Giantin in Patients with Alcoholic Liver Cirrhosis

In this current study, we are examining the link between giantin and EtOH-induced Golgi disorganization. First, we wanted to establish a relevance for these effects in the human condition, so we analyzed Golgi morphology in liver tissue samples obtained from patients with normal liver function and patients with alcoholic liver cirrhosis. Contrary to the normal cells, most of the cells from alcoholic samples exhibit remarkably altered Golgi structure (Figure 1A,B). Second, the level of giantin dimer in the tissue lysate of these patients was significantly lower than in control samples (Figure 1C). The detection of giantin dimer was performed as previously reported [15]; briefly, the lysis of cells was performed under high (5%) and low (1%) β-mercaptoethanol (β-ME). By lowering β-ME level from 5% to 1%, more giantin dimer was detected in samples, confirming that the dimer is formed by a disulfide bond [13,14]. These data fit well with our recent in vitro and in vivo observations, indicating that alcoholic liver injury is associated with the loss of giantin dimer and significant alterations of Golgi morphology [16]. 

### 3.2. Post-Alcohol Golgi Recovery in the Rat Model

We found recently that in hepatocytes from the alcohol-fed rats, alteration of Golgi was associated with the impaired trafficking of ASGP-R to the cell surface [16]. This echoes our previous observations that chronic EtOH administration impairs liver receptor-mediated endocytosis, as in, for example, the uptake of asialoorosomucoid (ASOR) by ASGP-R [43]. These effects were identified after as early as one week of EtOH feeding and were clearly established after four–six weeks of feeding. Of importance to our current study is that the impaired endocytosis was quickly restored (by seven days) upon refeeding by control diet [49]. This phenomenon prompted us to also examine post-alcohol Golgi recovery in vivo. To do this, we analyzed Golgi morphology in rats fed with (a) control diet, (b) EtOH containing (36% of calories) Lieber–DeCarli diet for periods of five weeks, and (c) EtOH diet followed by 10 days feeding with the control diet. In control rat hepatocytes, ASGP-R was distributed in Golgi, cytoplasm, and at the periphery of the cell; however, as we have shown before, in hepatocytes from EtOH-fed rats, the cytoplasm was highly vacuolated, and ASGP-R was accumulated in the fragmented Golgi (Figure 2A,B,D) [16]. Of note, in recovered hepatocytes, the number of vacuoles was essentially reduced, Golgi appeared more compact and juxtanuclear, and multiple ASGP-R positive punctae were detected again at the cell’s periphery (Figure 2C,D). Golgi recovery was importantly accompanied by giantin re-dimerization and partial restoration of ASGP-R trafficking to the cell surface, as indicated by W-B of both cell lysate and plasma membrane (PM) fractions isolated from all three categories of rat hepatocytes (Figure 2E). Importantly, in the same animal studies, the alcohol-induced elevated level of serum alanine aminotransferase (ALT) was returned to the normal after refeeding [50] implying that restoration of Golgi and trafficking of hepatic proteins coincides with the repair of a damaged liver. 

### 3.3. Post-Ethanol Recovery of Golgi in Giantin-Depletes Cells

To examine the precise role of giantin in post-alcohol recovery, we monitored the Golgi morphology in VA-13 cells after EtOH withdrawal in presence of giantin siRNAs. As shown in Figure 3A and in agreement with our previous observation of these cells [16], Golgi morphology (stained by GM130) in EtOH-treated cells (35 mM EtOH for 72 h) looks predominantly disorganized, which returns to the classical perinuclear position when cells recovered after EtOH under normal growing conditions for another 48 h. As we showed previously and present here, giantin knockdown (KD) has no significant impact on the perinuclear position of Golgi morphology [22,24]. However, in cells lacking giantin, post-alcohol Golgi failed to recollect membranes into the organized structure (Figure 3A,B). The quantitative analysis indicates that cells experiencing a deficiency in giantin demonstrate the identical rate of Golgi recovery as EtOH-treated cells (Figure 3C). 

### 3.4. Post-Alcohol Recovery of Golgi and Trafficking of Hepatic Proteins Requires Non-Muscle Myosin IIB and Rab6a

Recently we found that EtOH-induced Golgi disorganization is governed by motor protein non-muscle myosin IIA (NMIIA) [51]. Additionally, another non-muscle myosin isoform, NMIIB, is involved in exocytosis and in vesicular trafficking at the *trans*-Golgi and *trans*-Golgi network [52,53], and both NMIIA and NMIIB can be bound to F-actin. It is interesting to note that although NMIIA and NMIIB share many biochemical features, they have been found on different membrane domains and ascribed to distinct functions, such as cytokinesis and cell motility [54,55,56]. In addition, NMIIA and NMIIB remain tightly bound for different lengths of time to F-actin during the ATPase cycle [57,58]. In other words, while the complex of NMIIA and F-actin is the short-term event, the binding of NMIIB to F-actin is prolonged, suggesting NMIIB as a motor protein for generation of sustained tension [59,60]. In light of these facts, we hypothesize that NMIIA and NMIIB may play a diagonally different role in Golgi morphology. To test this, we performed siRNA depletion of both NMIIA and NMIIB in VA-13 cells. We then measured the perimeter of Golgi, using ImageJ, taking into account only membranous-specific giantin staining. As we predicted, NMIIA KD has no significant impact on Golgi [51], because Golgi size was comparable to cells transfected with scramble siRNAs (Figure 4A–C). However, in NMIIB-depleted cells, the perimeter of Golgi was significantly enlarged (Figure 4A–C). The data imply that under normal conditions, NMIIB controls Golgi integrity and could be essential for the recovery of Golgi.

These surprising findings raise the question of what additional players are involved in Golgi biogenesis. We have recently shown that giantin and NMIIA can compete for the Rab6a GTPase: in cells treated with EtOH, giantin de-dimerization was accompanied by the loss of its link to Rab6a, which results in a complex between Rab6a and NMIIA; the latter creates a force for EtOH-induced Golgi disassembly [51]. Therefore, it is logical to suspect that Rab6a may assist giantin in its re-dimerization.

With this background, we next investigated the distribution of ASGP-R in post-EtOH VA-13 cells in absence of either giantin, NMIIB or Rab6a. As anticipated, in control VA-13 cells, the ASGP-R signal was detected at the cell’s periphery (Figure 5A, Ctrl, white arrowheads) and Golgi. Treatment with 35 mM EtOH for 72 h induces Golgi fragmentation and reduces the ASGP-R IF peripheral signal (Figure 5A, EtOH). By contrast, when cells recovered after EtOH under normal growing conditions for another 48 h, Golgi was restored and ASGP-R signal was consistent with the control sample (Figure 5A, EtOH/Rec, white arrowheads). Nevertheless, post-EtOH recovery of ASGP-R peripheral staining was blocked in giantin-depleted cells (Figure 5A, Giantin KD + EtOH/Rec), which, as we shown before, are failed to recover Golgi (Figure 3). Similarly, we found no significant restoration of Golgi and ASGP-R in NMIIB- or Rab6a-depleted cells, nor in cells transfected with dominant negative (GDP-bound) Rab6a(T27N) (Figure 5A–C). Notably, we could not detect significant changes in perinuclear position of Golgi in control VA-13 cells treated with either Rab6a siRNAs or Rab6a(T27N), suggesting that disorganization of Golgi can be ascribed to EtOH effect only [61]. Thus, these data imply that post-EtOH Golgi recovery requires giantin, and it is mediated by the GTPase activity of Rab6a and action of NMIIB. 

As a way to evaluate the trafficking of liver-specific proteins to the cell surface, in addition to ASGP-R, we also measured by W-B the PM content of the PIGR and transferrin. As shown in Figure 5D, the intensity of bands of all three proteins was reduced in EtOH-treated cells, but in EtOH-recovered cells was very close to the value we saw in control cells. Of note, cells recovered from EtOH under giantin, NMIIB or Rab6a depletion, or transfected with Rab6a(T27N), express ASGP-R, PIGR, and transferrin at the level of EtOH-treated cells (Figure 5D). Next, to examine the level of complete glycosylation, we employed the *Sambucus nigra* agglutinin (SNA lectin) that specifically binds to sialic acid attached to terminal galactose in α-2,6 and to a lesser degree, α-2,3 linkage [62]. Predictably, in control VA-13 cells, the PM-associated ASGP-R bears sialylated *N*-glycans, indicating that this protein underwent full posttranslational modification [63]; however, in EtOH-treated cells, the expression of ASGP-R carrying sialylated *N*-glycans has been compromised [22,64]. In the meantime, the renaissance of Golgi in EtOH-recovered cells was importantly accompanied by the recovery of glycosylation (Figure 5E). Notably, sialylation of ASGP-R was reduced in cells lacking giantin and recovered from EtOH. 

## 4. Discussion

Our data confirm that Rab6a plays a dual role in the function of Golgi. On one hand, the complex between Rab6a and NMIIA has been shown to be involved in EtOH-induced Golgi remodeling and the extension of Golgi tubules to the ER [15,51,65,66]. On the other hand, as we show here, in cells lacking active Rab6a, Golgi could not revert to the juxtanuclear position during alcohol abstinence. Since both Rab6a and giantin exist in the dimer form, and Golgi restoration coincides with giantin dimerization, it is logical to assume that Golgi de novo formation is associated with simultaneous cross-bridging dimerization of both giantin and Rab6a. This scenario is based on (a) our previous observation that giantin and NMIIA compete for Rab6a [51]; (b) KD of Rab6a drastically reduces the amount of giantin [15]; and (c) preliminary data from the cells recovered after Brefeldin A treatment [67], which fits well with the data presented here. 

If giantin plays a leading role in the maturation of Golgi membranes, then, what is the role for GM130 and GRASP65, which have been shown to be essential for the lateral cisternal fusion during Golgi assembly and efficient glycosylation [68]? Notably, these data were not totally reproduced by another group, which demonstrates that the function of GM130 is rather necessary for the incorporation of the ER-emanating tubulovesicular carriers into the *cis*-Golgi stacks [69]. While we do not rule out that under normal circumstances, GM130 and GRASP65 may play a certain role in both events, we could not find any pieces of evidence supporting the critical role of these proteins in Golgi recovery after alcohol treatment. Previously we found that the expression and Golgi localization of GM130 and GRASP65 are not affected in EtOH-treated VA-13 cells [16], and here, neither GRASP65 nor GM130 siRNA-mediated KD prevents post-EtOH Golgi recovery in VA-13 cells [61]. The complex GM130–GRASP65 may serve as an alternative docking site for some Golgi residential proteins [24]; however, given that giantin is exclusively required for Golgi localization of important enzymes [15,24,25], glycosylation in EtOH-treated cells is significantly impaired. Therefore, post-alcohol recovery of giantin is required for proper glycosylation of hepatic proteins and their subsequent delivery to their working sites. In the meantime, it is known that under-glycosylated transferrin can be secreted into the bloodstream [27]. This poses the intriguing question whether compensatory mechanisms exist for abnormal glycoproteins to bypass fragmented Golgi en route to the cell surface. The most likely pathway is the direct ER-PM contact sites that are observed in different cells, including hepatocytes [70,71]. In alcohol-treated cells, this may serve as the alternative way to relieve the ER-stress, in addition to the unfolded protein response [72], however this possibility requires further assessment. 

Here, for the first time, we observed the differential impact of depletion of NMIIA and NMIIB on Golgi. It is known that cells pretreated with NMIIA siRNA or its inhibitors demonstrate a significant delay in Brefeldin A-induced Golgi disorganization [73,74]. We have also shown that NMIIA is tethered to Golgi membranes under heat shock, or inhibition of heat shock proteins (HSPs), and depletion of a β-COP subunit of COPI vesicles [3,48,51]. Finally, cooperation between NMIIA and Rab6a is essential for EtOH-induced Golgi fragmentation [51]. Thus, NMIIA is the master of Golgi remodeling; however, under normal conditions, its depletion has no visual impact on Golgi morphology, while KD of NMIIB leads to Golgi enlargement. It appears that compact structure and perinuclear position of Golgi are determined inter alia by a dynamic equilibrium between NMIIA that operate in its breakdown and NMIIB that is responsible for its maintenance. Indeed, here we found that depletion of NMIIB prevents the restoration of Golgi, indicating that the formation of the compact Golgi structure is somehow controlled by this motor protein. At this point, the mechanism of NMIIB interference with Golgi remains enigmatic. Analogously to NMIIA, NMIIB could be recruited to the Golgi by some local GTPase and tethered to the membranes via interaction with Golgi residential proteins [74]. Our preliminary data indicate that the possible candidate is *trans*-Golgi localized Rab3D [75], but the direct link between Rab3D and NMIIB needs further rigorous investigation. 

In sum, our results confirm the critical role for giantin, Rab6a, and NMIIB in the post-alcohol recovery of Golgi. We believe that the restoration of Golgi is a much more complicated event and requires the active involvement of other players. However, these three proteins appear to be the key regulators of fusion of the nascent Golgi membranes, which is the critical step in Golgi biogenesis. Our data support observations of chronic alcohol consumption that indicate the ability of hepatocytes to prompt recovery during alcohol abstinence. However, some parameters require more time to return to the original numbers. For instance, rats fed with Lieber–De Carli diet for three weeks demonstrate the inability to increase the level of Mg^2+^ in the extracellular compartment, and it takes ten days of EtOH withdrawal to restore Mg^2+^ [76]. Intriguingly, recent observation of alcohol withdrawal in patients with ALD indicates that despite the level of aspartate aminotransferase (AST) returning to normal during alcohol detoxification, the expression of the apoptotic marker, caspase-cleaved keratin-18 fragment, in the serum of these patients still remains high [77]. These data suggest that the evaluation of liver cells reparation requires additional parameters. Here, for the first time, we show that the retrieval of critical hepatic proteins depends on the restoration of compact Golgi morphology and its perinuclear position. Therefore, in the biopsy samples from patients with alcoholic hepatitis, in addition to the ballooning and Mallory–Denk bodies [78], the morphology of Golgi would be another critical histological aspect to monitor. Continuous structural disorganization of Golgi upon heavy alcohol drinking exhausts its recovery mechanism, potentiates ER-stress in hepatocytes and induces apoptosis [79], which, in turn, results in the manifestation of ALD. 

## 5. Conclusions

The ability of Golgi apparatus to recover after severe attacks is unique and could play a significant role in cellular homeostasis. Here, we describe the role of the largest golgin, giantin, in the maintenance of Golgi stability. Our results clearly indicate that giantin is required for post-alcohol restoration of Golgi, and the latter is a prerequisite for successful targeting of hepatic proteins to the cell surface. Moreover, we found that the reversal of Golgi to normal morphology also depends on the activity of Rab6a GTPase and the action of NMIIB. 

## Figures and Tables

**Figure 1 biomolecules-08-00150-f001:**
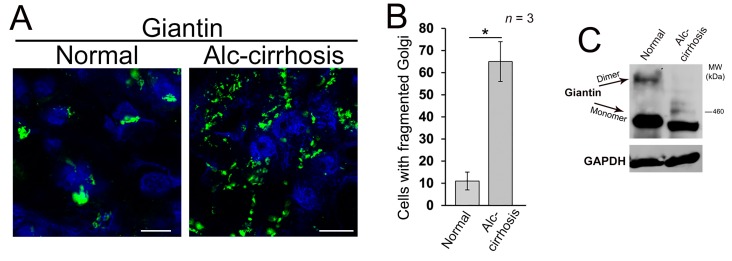
Alcohol-induced Golgi disorganization in patients with alcoholic liver cirrhosis. (**A**) Confocal immunofluorescence images of giantin in the liver tissue samples obtained from patients with normal liver function and patients with alcoholic liver cirrhosis; bars, 5 μm; (**B**) Quantification of cells with fragmented Golgi; *n* = 3 samples for each group. Results are expressed as a mean ± standard deviation (SD); * *p* < 0.001; (**C**) Giantin Western blot (W-B) of lysates from samples described in (**A**). Lysates were normalized to glyceraldehyde-3-phosphate dehydrogenase (GAPDH).

**Figure 2 biomolecules-08-00150-f002:**
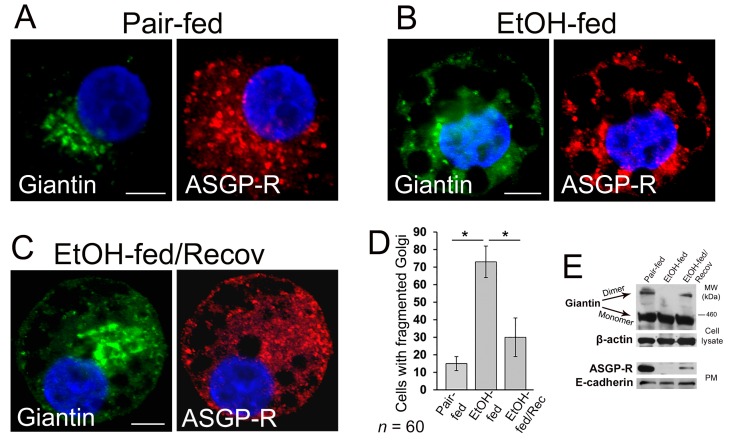
Post-alcohol Golgi biogenesis in vivo. (**A**–**C**) Giantin and asialoglycoprotein receptor (ASGP-R) immunostaining in hepatocytes obtained from rats: pair-fed (**A**), ethanol (EtOH)-fed (**B**), and EtOH-fed followed by the recovery; bars, 5 μm (**C**); (**D**) Quantification of cells with fragmented Golgi in cells from (**A**–**C**); *n* = 60 cells from two independent experiments, results are expressed as a mean ± SD; * *p* < 0.001; (**E**) Top panel: giantin W-B of lysates from cells described in (**A**–**C**); β-actin is a loading control. Low panel: ASGP-R W-B of plasma membrane fractions from cells described in (**A**–**C**); samples were normalized to E-cadherin.

**Figure 3 biomolecules-08-00150-f003:**
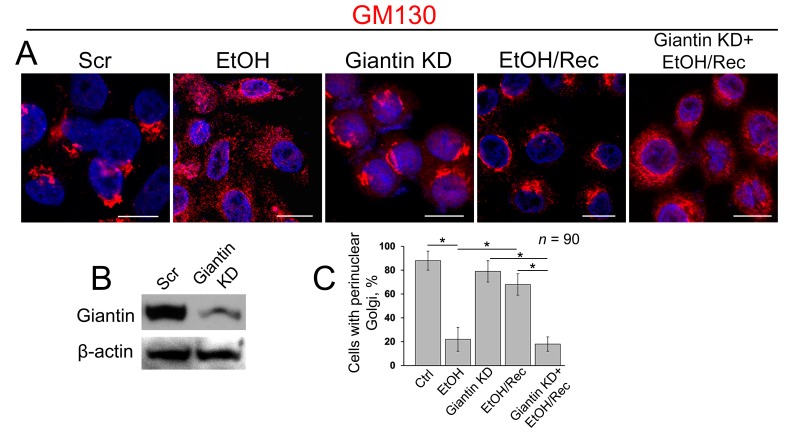
Giantin depletion prevents Golgi restoration in cells recovered from EtOH. (**A**) Confocal immunofluorescence images of Golgi (GM130) in VA-13 cells: treated with scramble small interfering RNAs (siRNA), treated with 35 mM EtOH for 72 h, giantin siRNAs, EtOH-treated followed by recovery for 48 h, or recovered in the presence of giantin siRNAs; bars, 10 μm; (**B**) Giantin W-B of lysates of VA-13 cells treated with corresponding siRNAs; β-actin was a loading control; (**C**) Quantification of cells with perinuclear Golgi in cells presented in (**A**); *n* = 90 cells from three independent experiments, results expressed as a mean ± SD; * *p* < 0.001.

**Figure 4 biomolecules-08-00150-f004:**
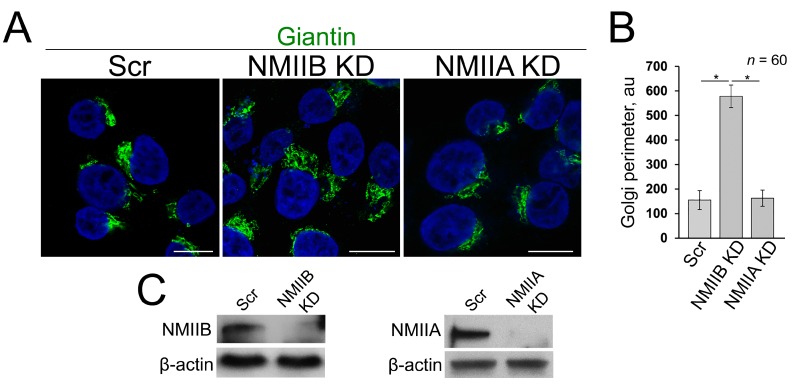
(**A**) Confocal immunofluorescence images of Golgi (giantin) in VA-13 cells: treated with scramble, NMIIB or NMIIA siRNAs; bars, 10 μm; (**B**) Quantification of Golgi perimeter from cells presented in (**A**); *n* = 60 cells from three independent experiments, results expressed as a mean ± SD; * *p* < 0.001; (**C**) NMIIB (left panel) and NMIIA (right panel) W-B of lysates of VA-13 cells treated with corresponding siRNAs; β-actin was a loading control.

**Figure 5 biomolecules-08-00150-f005:**
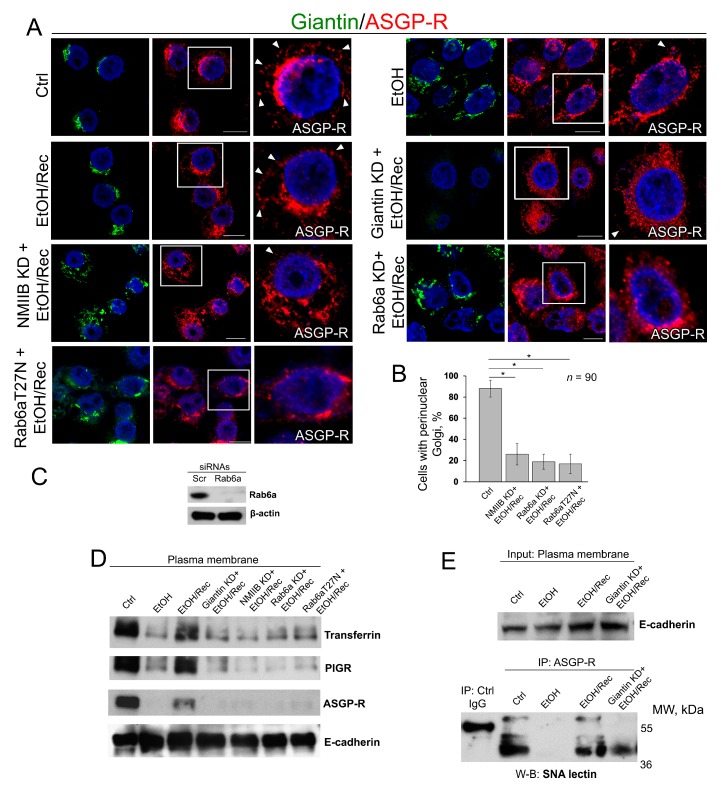
Giantin, Rab6a, and NMIIB are required for in vitro post-EtOH recovery of Golgi. (**A**) Confocal immunofluorescence images of giantin and ASGP-R in VA-13 cells: control, EtOH-treated cells, EtOH-treated cells and transfected with scramble, giantin, NMIIB, Rab6a siRNAs, and dominant negative (GDP-bound) Rab6a (T27N) followed by recovery. White boxes are enlarged pictures of ASGP-R presented at the right side. Arrowheads indicate ASGP-R punctae distributed at the periphery of cells. All confocal images are acquired with the same imaging parameters; bars, 10 μm; (**B**) Quantification of cells with perinuclear Golgi for indicated cells (assessment of EtOH, EtOH/Rec and giantin KD + EtOH/Rec cells was presented in Figure 3C); *n* = 90 cells from three independent experiments, results expressed as a mean ± SD; * *p* < 0.001; (**C**) Rab6a W-B of lysates of VA-13 cells treated with corresponding siRNAs; β-actin was a loading control; (**D**) Transferrin, PIGR, and ASGP-R W-B of plasma membrane fractions isolated from VA-13 cells presented in (**A**); samples were normalized to E-cadherin; (**E**) SNA lectin W-B of ASGP-R-IP from the plasma membrane fractions isolated from VA-13 cells: control, EtOH-treated, and recovered from EtOH in absence or presence of giantin siRNAs. The input was normalized to the E-cadherin.

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
