# Peer review of "Giantin Is Required for Post-Alcohol Recovery of Golgi in Liver Cells"

_biomolecules, 2018, doi:10.3390/biom8040150_

Round 1

Reviewer 1 Report

The study by Casey et al., entitled “The role of Golgi morphology in post-alcohol recovery of hepatocytes: observations in cellular and animal model” investigates the mechanisms of Golgi recovery during ethanol withdrawal. First, the authors demonstrated alterations of Golgi morphology in the liver tissues obtained from the alcoholic cirrhosis patients, and further examined the mechanisms of Golgi recovery from ethanol-mediated alterations  in VA-13 cell line and in primary hepatocytes isolated from experimental rats. Overall, it is a high quality molecular and cellular level study demonstrating that Golgi post-alcohol recovery involved modulation of several proteins, including giantin, Rab6a, and NMIIB. 

There are several minor concerns the authors should consider in order to improve a quality of the manuscript: 

-              The title of the manuscript is very generic (would better fit for a review article), and does not reflect the main findings of the study. 

-              Introduction is too long and difficult to follow. 

-              The authors should check the reference list. It seems that some references do not necessarily match the reference in the text.

-              Result section should be divided into several sub-sections with dedicated titles 

-              What was NMIIA and NMIIB siRNA efficiency and how it was controlled? 

-              The Golgi alterations should be connected to liver pathology in animal models

-              The authors should discuss the results obtained on human liver tissue samples. 

Author Response

               Thanks a lot for your effort in reviewing this manuscript and the positive appraisal of our story. Below are our point-by-point responses to the comments:

 The title is changed, the Introduction is shortened and simplified, and References are checked carefully. Also, the Results section is divided into sub-sections with the appropriate titles. The efficiency of NMIIA and NMIIB siRNA-mediated knockdown is presented in Fig. 3B. These siRNAs have been tested by our group many times. Also, we made appropriate changes regarding discussion of Golgi morphology in human samples and in animal studies. These changes are highlighted by a yellow marker.  

Reviewer 2 Report

This manuscript was well written. I just have minor questions:

(1) Line 222, Figure, what are the levels of ALT in the animals after alcohol feeding and post-alcohol recovery?

(2) Line 363, please change "that responsible" to " that are responsible".

Author Response

Thanks a lot for the positive appraisal of our story. Below are our point-by-point responses to the comments:

Question: Line 222, Figure, what are the levels of ALT in the animals after alcohol feeding and post-alcohol recovery?

Answer: the alcohol-induced elevated level of serum alanine aminotransferase (ALT) was returned to the normal after refeeding, implying that restoration of Golgi and trafficking of hepatic proteins coincides with the repair of a damaged liver. We did not include these data here since we are going to include it to our next story, which is more clinically relevant.

Question: (2) Line 363, please change "that responsible" to " that are responsible".

Answer: Thanks, it is fixed.

Reviewer 3 Report

The manuscript reports the effect of alcohol on Golgi morphology and the mechanisms involved in disassembly and re-assembly in the setting of alcohol exposure and withdraw, respectively. The authors demonstrated that giantin dimerization status correlates well with for alcohol-induced Golgi disassembly (de-dimerization) and after alcohol recovery-associated Golgi re-assembly (re-dimerization). Cell culture study showed that giantin, Rab6a and non-muscle myosin IIB contributes to post-ethanol recovery of Golgi. The Golgi apparatus plays an important role in cellular protein homeostasis, but the effects of alcohol on Golgi is poorly studied. This study presents interesting data in defining factors involved Golgi assembly in the setting of alcohol exposure and recovery. 

Below are concerns for the authors to address.

 1.      Figure 3 shows that giantin KD prevents Golgi restoration in hepatoma cells recovered after ethanol exposure. The data suggest that giantin critically mediates Golgi biogenesis. While the data are solid, it is very interesting to know whether giantin KD also prevents the recovery from alcohol-induced cell injury. Did the authors get chance to measure cell injury such as LDH release with this cell culture model?

Author Response

Thanks a lot for the positive appraisal of our story. Below are our point-by-point responses to the comments:

Question:   Figure 3 shows that giantin KD prevents Golgi restoration in hepatoma cells recovered after ethanol exposure. The data suggest that giantin critically mediates Golgi biogenesis. While the data are solid, it is very interesting to know whether giantin KD also prevents the recovery from alcohol-induced cell injury. Did the authors get chance to measure cell injury such as LDH release with this cell culture model?

Answer: Thanks for this comment, unfortunately, we did not do this, but we are working on the next story to present more clinically relevant data regarding recovery of hepatocytes after alcohol withdrawal. However, we mentioned in this story that in the animal study, the alcohol-induced elevated level of serum alanine aminotransferase (ALT) was returned to the normal after refeeding, implying that restoration of Golgi and trafficking of hepatic proteins coincides with the repair of a damaged liver. 

Reviewer 4 Report

This is a very interesting analysis of the recovery of Golgi structure post-ethanol-induced disruption of the Golgi.  The authors identify specific proteins required for the recovery, revealing basic mechanisms of Golgi biology.  In general this is well written and well designed and the figures are clear.

One suggestion: it would be helpful to show data on the golgi structure in the Gigantin knock-down in control cells and also Ethanol treated cells before recovery. 

Author Response

Thanks a lot for the positive appraisal of our story. Below are our point-by-point responses to the comments:

Question:   This is a very interesting analysis of the recovery of Golgi structure post-ethanol-induced disruption of the Golgi.  The authors identify specific proteins required for the recovery, revealing basic mechanisms of Golgi biology.  In general this is well written and well designed and the figures are clear. One suggestion: it would be helpful to show data on the golgi structure in the Giantin knock-down in control cells and also Ethanol treated cells before recovery. 

Answer: Thanks for this comment, the picture of cells with giantin KD and EtOH-treated cells are presented in the updated Figure 3.